# Novel Insights into Plant Genome Evolution and Adaptation as Revealed through Transposable Elements and Non-Coding RNAs in Conifers

**DOI:** 10.3390/genes10030228

**Published:** 2019-03-18

**Authors:** Yang Liu, Yousry A. El-Kassaby

**Affiliations:** Department of Forest and Conservation Sciences, The University of British Columbia, 2424 Main Mall, Vancouver, BC V6T 1Z4, Canada

**Keywords:** evolutionary genomics, adaptation, adaptive evolution, transposable elements, non-coding RNAs, genome purging, conifers

## Abstract

Plant genomes are punctuated by repeated bouts of proliferation of transposable elements (TEs), and these mobile bursts are followed by silencing and decay of most of the newly inserted elements. As such, plant genomes reflect TE-related genome expansion and shrinkage. In general, these genome activities involve two mechanisms: small RNA-mediated epigenetic repression and long-term mutational decay and deletion, that is, genome-purging. Furthermore, the spatial relationships between TE insertions and genes are an important force in shaping gene regulatory networks, their downstream metabolic and physiological outputs, and thus their phenotypes. Such cascading regulations finally set up a fitness differential among individuals. This brief review demonstrates factual evidence that unifies most updated conceptual frameworks covering genome size, architecture, epigenetic reprogramming, and gene expression. It aims to give an overview of the impact that TEs may have on genome and adaptive evolution and to provide novel insights into addressing possible causes and consequences of intimidating genome sizes (20–30 Gb) in a taxonomic group, conifers.

## 1. Introduction

Gymnosperms today comprise a little more than 1000 species that are two to three orders of magnitude lower than *c*. 352,000 species of extant angiosperms [1]. Yet, gymnosperms have a long and extensive fossil record that dates back to the Carboniferous (*c*. 290 million years ago (mya)) [2,3]. By contrast, morphologically recognizable fossil angiosperms first appeared more recently (in the lower Cretaceous; *c*. 130 mya), and fossil evidence of their rise to ecological dominance emerged in the mid-Cretaceous (*c*. 100 mya) [4,5]. The sudden appearance and rapid diversification of angiosperms in the fossil records were, to C. Darwin, a “perplexing phenomenon” and an “abominable mystery” (letter from Darwin to Hooker 1887; [6]). In the ancient and widespread plant lineages of gymnosperms, two-thirds are conifers (Coniferales or Pinophyta), mainly including Pinaceae and Cupressophytes of 546–615 species [7,8]. This lineage plays an important role in global carbon, nutrient, and atmospheric cycles and is of great ecological and economic importance worldwide. The intimidating genome sizes of conifers have been a major bottleneck, constraining the exploration of the astonishing rise of angiosperms from gymnosperms at the molecular level. However, recent years have witnessed fascinating strides in our understanding of the genome evolution and adaptation in conifer species, owing to next-generation sequencing technologies and interdisciplinary developments, such as bioinformatics, quantitative and population genetics, and evolutionary biology. In the past decade, complete draft assemblies have been obtained for four conifer genomes: Norway spruce (*Picea abies*; [9]), white spruce (*Picea glauca*; [10,11]), loblolly pine (*Pinus taeda*; [12,13,14]), and sugar pine (*Pinus lambertiana*; [15]). More conifer genome sequencing is underway (e.g., *Pseudotsuga menziesii* and *Larix sibirica*; http://pinegenome.org/). From a genomic evolutionary perspective, the different propensities for genome expansion through polyploidy and/or repeat amplification vs. genome contraction through epigenetics suppression and various recombination-based mechanisms have led to contrasting genome sizes of organisms. For the genome of *Picea abies* and *Pinus taeda*, high proportion (*c*. 60%) of long-terminal repeat retrotransposons (LTR-RTs) [9,12,13] and unique non-coding RNA (ncRNA) generation features [16,17] have shed novel insights into genome evolution and adaptation in conifers. Here, we review some of the most updated examples on the roles of transposable elements (TEs) in plant genome evolution and adaptation through epigenetics mechanisms, whereby we discuss how TEs and ncRNAs (and DNA methylation) dynamics contribute to conifer genomic and adaptive evolution.

## 2. Transposable Elements: A Source of Genetic Innovation

TEs are stretches of DNA sequences that can move and amplify their copy number within a host genome [18,19]. TEs are a major source of genomic mutations [20], and TE insertions provide a potent mutagenic mechanism for the evolution of new genes and their functionalities [21,22,23,24]. For instance, the Ac/Ds family insertion into a starch synthase gene created Mendel’s wrinkled peas [25], a disruptive insertion of a phytochrome A paralog promoted adaptation to high latitudes [26,27], and the hopscotch insertion helped shape the genomic architecture of the modern maize plant [28]. TEs also provide the raw material from which novel regulatory sequences are derived [22,29], such as promoters and enhancers [30,31]. Such changes in *cis* regulatory element change the structure and the regulatory and/or epigenetic environment, leading to modified gene expression (see review [32]). In addition, TEs affect the genome architecture by facilitating chromosomal sequence rearrangements due to their potential to arouse chromosomal mutations, for example, in maize [33]. TE-mediated genome rearrangement has been generalized throughout all organisms, including plant species [34,35].

Studies using *Drosophila melanogaster* lines have demonstrated that fitness is negatively correlated with TE numbers [36,37]. In plants, a positive correlation between genome size and the extinction probability has been documented [38,39,40]. TEs often proliferate faster than they can be removed, thus contributing to genome growth or even genome obesity [41,42]. Generally, plant genome size variation is mainly due to polyploidization and TE proliferation and/or deletion; thus, variable TEs insertion and deletion rates constitute an essential driving force in genome size evolution [34]. Recent advances in our understanding of the centrality of TEs to genome size and genic evolution provide insights into the mechanistic underpinnings of biased fractionation and polyploidy events [21,22,42,43], depicted in Figure 1. Biased fractionation may result from changes in epigenetic landscape near genes mediated by TEs. These epigenetic changes result in unequal gene expression between duplicates, establishing different fitness, which leads to biased gene loss with respect to ancestral genomes, typically termed “biased fractionation” [44]. It has been proposed that the differential loss of ancestral genome is explained by two observations [45]. First, homologs of retained duplicated genes in the most highly fractionated genome are often expressed at lower levels than their counterparts in the more intact genome [44,46]. Second, epigenetically silenced TEs are often more physically adjacent to the homolog with lower expression, indicative of a positive-effect repression of gene transcription [47,48,49].

Moreover, some TEs fold into stem-loop secondary structures and thus potentially contribute to the formation of ncRNAs [50,51]. Because TEs serve as an important resource for ncRNA generation, biased fractionation can lead to quantitative and qualitative mismatches between silencing machineries, as described in [30,52]. These regulatory mismatches likely give rise to various perturbations in the ncRNA populations (e.g., production of novel ncRNAs), with corresponding effects on gene expression, which has cascading influences on individual fitness.

## 3. Transposable Elements–Epigenetics Components Interplay: An Evolutionary Force in Adaptation

In the past decade, major advances in understanding the molecular mechanisms underpinning adaptation (e.g., responses to stress and developmental process) have been made [53], notably by highlighting the relevance of two environment-sensitive molecular elements: TEs and epigenetic components (e.g., [54,55]). While TEs provide agents of adaptive evolution and adaptation by creating genetic variants and diversity [56,57] (or see review [58]), epigenetic mechanisms play regulatory roles by the alteration of epigenetic landscapes (see review [59,60]). Epigenetic components comprise a molecular network that can affect complex traits—sometimes transmitted across generations—in the absence of genetic variation [61,62,63]. For instance, patterns of DNA methylation can persist across generations and produce heritable phenotypic changes [64,65]. Cortijo et al. [65] demonstrated that specific heritable methylation patterns in *Arabidopsis thaliana* experimental strains accounted for 60%–90% of the heritability for flowering time and primary root length. Interestingly, TEs and epigenetic components are intimately connected through the environment, potentially amplifying their actions on phenotypes and genotypes [54,66,67,68], thus playing a central role in an organism’s adaptation [69,70].

TE activities can be triggered by environmental cues, accelerate mutation rates, and rewire regulatory networks [71,72]. As first put forth by McClintock [73], TEs comprise a significant adaptive response of the genome to environmental challenges. In particular, stress may stimulate bursts of TE activity [74,75], as may hybridization and polyploidy [35,76,77]. Stressors triggering TE activities have been instantiated in plants [42,78]. For instance, thermal stress induces genome-wide modifications of methylation patterns in the grass *Leymus chinensis* [79]; DNA methylation under drought stress in *Populus trichocarpa* occurs in TEs [80]. Field evidence also indicates that environmental conditions affect the epigenomes in wild populations [63]. For instance, invasive populations of Japanese knotweed (*Fallopia japonica*) established in various habitats in northeastern America exhibit massive epigenetic differentiation, largely exceeding the observed genetic differentiation, and some of the epigenetic patterns might respond to local habitat conditions [63]. Moreover, TEs can regulate gene expression by attracting repressive marks. In plants, TEs associated with long intergenic ncRNAs are tissue-specific and expressed under specific stressful conditions [81]. TEs can also modify gene expression in response to stress to which the elements are themselves responsive [82,83]. For instance, In some plants (e.g., wheat, tobacco, oat), specific DNA elements in the 5′ LTR-RTs were identified in relation to stress responses, such as phytohormones, heat, light, or salinity [84,85,86,87,88]. In maize and rice, stress-sensitive TEs were shown to be inserted in the flanking regions of some genes, inducing specific stress-responsive regulation of these genes [82,83]. These discoveries highlight that TEs and epigenetic components tightly interact through numerous pathways and suggest their joint implication in organisms’ responses to stress. Nonetheless, TEs do not always have a positive fitness effect under stress conditions (see review [89]), and a negative relationship between stress and TEs have been recently documented in plants [81,90]. As such, the TEs–stress relationship is complex, although evidence has shown that TEs are activated under stress in most studies performed so far, and TE insertions have been shown to have a beneficial effect in some cases (e.g., [91,92]).

A priori, epigenetic components play a major role in storing genetic information for particular phenotypes in a silent state as long as epigenetic marks are faithfully transmitted across generations, hence generating so-called hidden genetic variation [93]. Environmentally induced epigenetic changes can reveal hidden genetic variation, which provides a mechanism for rapid adaptation [94]. Moreover, under stress, TEs can be activated in somatic cells either directly or through the arousal of their epigenetic control, thus producing non-heritable phenotypic variation among somatic cells within an organism [95]. This mechanism, termed genetic mosaicism, can instantaneously generate adaptive phenotypic variation in response to stress, especially in long-lived organisms [95]. In addition, mobile ncRNAs may be involved in coordinated responses to stimuli perceived in different organs. For instance, small interfering RNAs (siRNAs) expressed in shoot cells (i.e., from photosynthetic organs) can move to root cells (i.e., water-providing organs) and modify DNA methylation profiles in the latter cells, hence providing a coordinating system between functional organs [96].

Epigenetics components are key in repressing TE activities, thus making TEs “well behaved” and protecting genome integrity against TEs’ disruptive mobility [54]. Many TEs are targeted by DNA methyltransferases, and the arousal of epigenetic silencing is often associated with the activation of TEs [97,98], thus partly accounting for the sensitivity of TEs to the environment [78]. Conversely, TEs contribute to the evolution of genetic and epigenetic regulatory networks [66,72]. ncRNAs are integral parts of the epigenetic regulatory machinery through interaction with enzymes involved in DNA methylation and histone modifications. Research has shown that several ncRNAs are encoded by TEs or by endogenous genes that are likely to be derived from TEs [99,100]. Thus, TEs are essential genomic components encoding elements involved in the epigenetic machinery [100]. Another aspect of ncRNAs encoded by TEs is that they can act to repress the proliferation of the TEs from which they originate through a sequence complementary match (i.e., TE silencing) [101]. Thus, epigenetically induced controls on the activities of ncRNAs constitute a pathway to modulate TE activities [100]. Collectively, ncRNAs are key elements in TEs–epigenetics components interactions.

In addition, recent research has also advanced our understanding of genome size evolution from an organismal adaptation perspective. Under stressful environments, TEs may be under purifying selection, resulting in a compact genome [102,103]. TE-mediated genome instability could be accelerated in the face of environmental insults, such as high salinity and strong UV exposure [104,105,106], leading to enhanced selection pressures on TEs. Even within the same species, different environmental exposures may lead to the variation of genome size and TE contents. For example, a study of independent adaptations to high altitude in *Zea mays* showed that genome size experiences parallel pressures from natural selection, causing a reduction in genome size with increasing altitude [107]. This study also highlighted that genome size evolution may be environmentally dependent and that correlated changes in genome size may be mediated through, for instance, flowering time [108,109,110].

## 4. Genome-Purging Mechanism: Another Evolutionary Force to Counterbalance Transposable Element Increase

The explosion of selfish genetic elements is typically thought to be controlled by unconditionally deleterious effects, such as the harmful effects of gene disruption, meiotic defects on pollen viability and seed set [111], and ectopic recombination events causing chromosomal rearrangements [112,113]. It was thought that plants were on a path to obesity through continual DNA bloating, but recent research supports that most plants actively purge DNAs [114]. Genome-purging mechanisms include illegitimate or unequal combination between LTRs and other types of deletions, which facilitates to counterbalance the ever increasing TEs [115]. Intra-strand homologous recombination between directly repeated LTRs deletes the sequences between LTRs, leaving solo-LTRs [116]. Analysis of solo-LTRs and comparison of internally deleted RTs has indicated that illegitimate intra-strand homologous recombination may be the driving force in maintaining slim genomes of *Arabidopsis* [117] and rice [118]. Recently inserted TEs are often removed from the genome (i.e., genome contraction), resulting in rapid genomic turnover. For example, the rice genome underwent several bursts of LTR-RTs during the last 5 million years but ultimately removed over half of the inserted LTR-RT DNA [119,120,121]. A three-fold increase in the genome size of diploid members of *Gossypium* is due to the accumulation of LTR-RTs over the past 5–10 mya [122].

Another mechanism that prevents genomes from uncontrolled expansion is through non-homologous end joining (NHEJ) after deletion-biased double-strand break (DSB), which leads to massive genomic restructuring by purging LTR-RTs in a small genome [123]. This pathway is frequently invoked as the possible cause of genome shrinkage (see review [124]). The genome of *Oryza brachyantha* is 60% smaller than its close relative cultivating species, *Oryza sativa*, and 50% of their size difference was found to be due to the amplification and deletion of recent LTR-RTs [123]. In *c*. 32,000 protein-coding genes of *O. brachyantha*, only 70% of them were in collinear positions as the rice genome [123]. It is therefore reasonable to argue that the low LTR-RT activity and massive internal deletions of the LTRs by NHEJ after DSB have shaped the current *O. brachyantha* genome. Likewise, NHEJ after DSB was found to be a common pathway in the genomic reduction of *A. thaliana* [125].

## 5. Transposable Elements in Conifer Genome Architecture

The conifer genomes are elusively large (20–30 Gb) [126] and abundant, and diverse RTs are the main component of non-genic portions [127,128]. It is estimated that 62% of *Pinus taeda* genome is composed of RTs, of which 70% are LTRs, mainly Pseudoviridae (also known as *Ty1/Copia* elements) and Metaviridae (*Ty3/Gypsy*) ([9,12,13]; also see summary of conifer TEs in a conifer TE database, ConTEdb [129] based on more than 0.4 million TEs of three sequenced conifer genomes). Similarly, the total representation of all TE classes was estimated as 69% in *P. abies* with similar types of LTRs [9]. The comparative analysis of LTR-RTs in conifers also indicated that the accumulation of retro-elements in conifer genomes is very ancient and has occurred over a very long timeframe spanning tens to hundreds of millions of years [9].

The large conifer genomes contain a huge wealth of divergent and ancient repeats, including RTs, which have most likely arisen through a combination of ongoing repeat amplification over long periods of evolutionary time [130,131], combined with a lack of efficient and/or slower rates of repeat elimination via recombination-based processes [9,132]. They have the lowest recombination rates relative to any eukaryotic lineages reported so far [132,133,134]. The possible reason for this is that, when genome size grows above a certain threshold, large tracts of heterochromatin are formed from stretches of repeats, which become “locked down” into highly condensed chromatin through epigenetic activities. This leads to the reduction of the potentially negative impact of TE activity and limits the accessibility of the recombination and hence the potential of recombination-based processes to eliminate DNA [66]. Surprisingly, LTR-RTs of the Norway spruce genome are mostly low-copy LTRs with 80% singletons [9]. The intimidating genome size seems to have resulted from a slow but steady accumulation of diverse LTR-RTs, which might be due to the lack of efficient elimination mechanisms, as evidenced by the fact that pine species also have low-copy LTRs [127].

In addition to TE characteristics, conifers share other common features, making them distinct from other spermatophytes. For instance, conifer genome sizes have a 5.5-fold difference (1C = 6.6 to 36 pg), and most conifers have a remarkable constancy in chromosome numbers of 2n = 24 [135]. Natural polyploids are exceedingly rare in conifers (and gymnosperms in general) [136,137], indicating that WGD is not an important evolutionary driving force in conifer genomes. Moreover, conifers have particularly low average linkage disequilibrium [138,139], which is often confined within genes [140]. These distinct conifer genomic features indicate largely independent evolution of individual genes [141], favoring efficient natural selection instead of genetic drift, leading to genetically based clines (see review [142]). Finally, conifer genes tend to accumulate long introns, with the largest introns spanning 60 kb in spruce [9] and 120 kb in pine [13]. One or a few long introns of several kb are found in conifer genes and conserved across species [143]. However, no evidence has shown that very long introns in conifers would reduce the level of gene expression [9,143]. This is in contrast to findings in mammalians, which report high expression in genes with shorter introns [144].

As outlined for the nine conifers most frequently distributed worldwide, the genome size variation is around two-fold (Figure 2A), comparable with the *c*. 5.5-fold variation within all conifers [135]. The genomes of the four sequenced conifers are significantly larger than those of non-conifer tree species (*p* = 0.0035; Figure 2B). However, the genome size variation seems to be independent of the number of protein-coding genes, and there is no significant difference in gene numbers between conifers and non-conifer trees (Figure 2C; also see [145]). Conifers have similar numbers of unigenes, less than 40,000 as in many non-conifer trees (*p* = 0.98; Figure 2C). Moreover, conifer genomes have more class I TEs compared with non-conifer trees (*p* = 0.005) but have very low content of class II TEs compared with non-conifer trees (*p* = 0.94; Figure 2D) and most angiosperms [9,143]. Nonetheless, a high ratio of class I TEs is not a distinct feature between conifers and angiosperms as class I TEs make up as high as 76% of maize genome, leaving another 10% for other types of TEs [146], and constitute over 65% of 17 Gb wheat genome [147]. In addition, variation in genome size and variation in chromosome number are not correlated in trees (Figure 2C and Appendix A), which is in agreement with findings in flowering plants in general [148].

## 6. Non-Coding RNA Features in Conifers

Excessive TEs can destroy the genome, and many organisms have developed diverse mechanisms to inhibit TE activities, including RNA-based silencing pathways [149,150]. Insights into the role of epigenetic pathways, including the siRNA-directed DNA methylation (RdDM) in plants, to suppress activity of TEs as well as other counterbalancing mechanisms is an active area of research [52]. Endogenous 24-nt-long ncRNAs are enriched in intergenic and repetitive genomic regions [151,152], and more generally, most plant ncRNA loci are mapped to intergenic segments [50]. The feature of exceptionally large amount of excessive genomic DNA in conifers implies a rich source of TE-derived ncRNAs, thus building a distinct landscape of ncRNA populations targeting TEs and genes involved in various biological processes. In particular, 24-nt-long ncRNAs act as “genome guardians”, providing multigenerational protections against invasive TEs. The lack of regulation of 24-nt-long ncRNAs, which occurs at late seed set in conifers, alters developmental programs and may lead to increased TE activities and exceptionally large genome size [16]. The trade-off between large amount of TEs and novel ncRNAs, which are originated from foldback TEs to restrain TE expansion, may steer the direction and speed of genome evolution.

In addition, genome-wide average DNA methylation levels are positively correlated with genome size and TE contents [153]. Conspicuous 24-nt-long ncRNAs [9,16] and a very high level of DNA methylation [153] uniquely yielded at the reproductive period in conifers. This pattern of ncRNA dynamics prompts interesting investigations in linking ncRNAs to methylation dynamics to unravel the roles of epigenetics in genomic and adaptive evolution.

## 7. Outstanding Questions

With updated knowledge of TEs and ncRNAs in mind and considering the genomic features of conifers and several incipient studies on conifer epigenetics, we put up the following four questions and hypotheses for further testing:To what extent do TEs contribute to conifer genetic variation and population divergence? Based on TE polymorphisms, what are TE dynamics in natural populations and how are these loci under selection?⇨Given the availability of reference genome and TE consensus sequences, computational approaches (e.g., [154]) provide a possibility of addressing this question.How does the landscape of ncRNAs and DNA methylation alter over time (e.g., reproduction vs. vegetative growth) in the region of TEs (or different types of TEs), genes, and introns of focus conifers?⇨We could additionally focus on polymorphic TE loci identified in the previous question and genes involved in known pathways to explore how epigenetic components work in synergy in those “hotspots”.What are the features of TEs–epigenetics components (ncRNA and DNA methylation) interactions when individuals are exposed to stressful vs. benign environmental conditions? How are these features associated with local adaptation?⇨By identifying polymorphic TEs among populations (e.g., [155]), we are able to infer whether different types of TEs have undergone differential expansion or contraction, hence playing a role in adaptive evolution. Then, contrasting these results with their DNA methylation landscape permits testing whether and how the two mechanisms jointly contribute to adaptation.Is there evidence supporting genome-purging mechanisms in conifers?⇨We could look for answer by, for example, estimating death rates of LTR-RTs (e.g., prediction by [156,157]) by counting solo-LTRs and truncated elements in sequenced conifers and their close relatives.

## 8. Closing Remarks

Upon the discovery of transposable elements in the 1950s, Barbara McClintock envisioned that these sequences might operate to control gene expression and play a major role in evolution. Her prophetic remarks are finally receiving growing empirical support after several paradigm shifts in the past decades. We now know that a large amount of so-call ”junk” or ”parasite” or ”selfish” DNAs in plant taxa, especially in conifers, are not true junks but ubiquitous and influential genetic elements, and this has posited a meaningful conundrum in evolutionary genetics and genomics. They are now considered as drivers in evolution and are the focus of numerous genomic studies. The activity of LTR-RTs is under the control of epigenetic suppressing mechanisms. Epigenetic regulations of the transposable elements are the first line of defense against uncontrolled transposable element proliferation. Also, genome-purging mechanisms have been adopted to counterbalance the genome size amplification. Thus, the current genome evolution of organisms may be driven by a long battle of repeat sequence amplification and genome-purging systems in which conifers should represent an outstanding node for comparative studies. In addition, TEs can generate a broad range of genetic variation in natural populations by interplaying with epigenetic components, highlighting the interactive roles of TEs, epigenetic components, and the environment in adaptive evolution and adaptation. Thus, TEs in conjunction with epigenetics landscapes (ncRNAs and DNA methylation) provide a novel avenue to unravel the molecular underpinnings of local adaptation in long-lived perennials, such as conifers.

## 9. Terminology

Biased fractionation: the unequal loss of genes from ancestral progenitor genomes, which is a frequent event after polyploidy in many lineages.Class I elements of TEs: retrotransposons, which use reverse transcriptase to copy an RNA genome into the host DNA (i.e., “copy and paste”); see a unified TE classification system [158,159] and illustration by Chénais et al. [24].Class II elements of TEs: DNA transposons; the DNA genome of the element itself serves as the template for transposition either by a “cut and paste” mechanism or using a rolling circle process.Long terminal repeats (LTRs): identical DNA sequence that can be repeated at the ends of retrotransposons.Non-coding RNAs (ncRNAs): randomly grouped into short (<200 nt) and long (>200 nt) types [160]. Members of short ncRNAs involved in plant transcriptional (indirect and low) and post-transcriptional (major) regulations have been well documented (e.g., review in [52]). These short ncRNAs chiefly consist of microRNAs (prevalence of 21 or 22 nt long in suppressing target mRNAs), heterochromatic small interfering RNAs (hc-siRNAs; 24 nt mediators in silencing DNA methylation and histone modifications), and trans-acting siRNAs (tasiRNAs or phasiRNAs; 22 (or 21) nt with a phased configuration, playing similar roles as microRNAs or other uncharacterized functions).Whole-genome duplication (WGD) or polyploidization: an event in which the entire genome of an organism is copied once or multiple times. A widely accepted hypothesis for WGD events is based on a hexaploidization of all eudicots (ancestral γ events), first put forth by Jaillon et al. [161].

## Figures and Tables

**Figure 1 genes-10-00228-f001:**
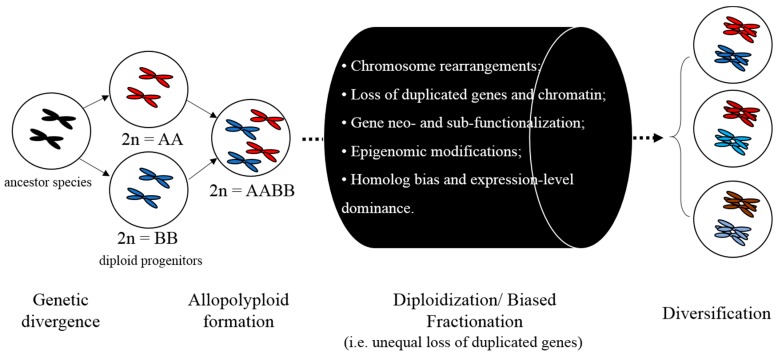
Conceptual illustration of key realizations in plant genome architecture. The merger of two diploids via hybridization and/or allopolyploidy has novel evolutionary consequences [21], enumerated in the black cylinder.

**Figure 2 genes-10-00228-f002:**
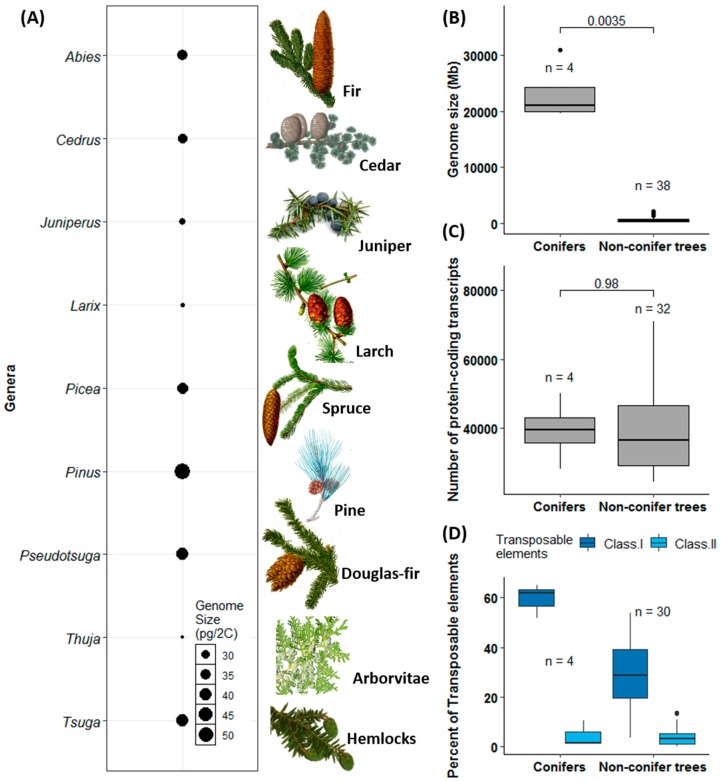
Comparison of genome size, protein-coding transcripts, and transposable elements between conifers and other sequenced tree species. In (**A**), estimated mean genome sizes of major conifer species is based on the Plant DNA C-values Database (http://data.kew.org/cvalues/) as of February 2019. All botanical illustrations are from the public domain (Note S1). In (**B**–**D**), the number of species (n) used in each calculation is different due to incomplete information available in original publications. The *p*-values based on *t*-tests are given on the top of each panel. *p*-values for the comparisons of class I and II TEs between conifers and non-conifer trees were 0.005 and 0.94, respectively. The data used for these plots were extracted from the publications listed in Appendix A.

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
