# Peer review of "Novel Insights into Plant Genome Evolution and Adaptation as Revealed through Transposable Elements and Non-Coding RNAs in Conifers"

_genes, 2019, doi:10.3390/genes10030228_

Round 1

Reviewer 1 Report

This review paper by Liu and El-Kassaby describes the role of transposable elements and non-coding RNA in the evolution of plant genomes, and especially conifers genome. This review is of great interest for readers in the field of plant science and/or genomic evolution. The manuscript is well written and easy to read; moreover, the citations chosen are appropriate and the list is quite complete. This original review gives an interesting and complementary point of view on evolutive mechanisms driven by Transposable Element from one hand and non-coding RNA from the other hand. I recommend its publications in GENES after a few minor changes.

Minor concerns:

1) Line 13, the term “saltational” is difficult for a non-specialist reader to understand and could be replaced by a more common term.

2) Line 73, I suggest to add reference [1] to the cited references, currently [22-24].

3)  About reference [1]; it seems that an author is missing, namely the second author A. Caruso. 

4) Line 134, I would like to suggest the authors adding an example of transposable element expression in condition of thermal stress. Namely, the LTR-retrotransposon Surcouf was shown to be overexpressed under thermal stress in the microalga Phaeodactylum tricornutum. Moreover, heat-shock response elements, stress response elements and CCAAT box (collectively referenced as HSEs) were identified in the promoter of the retrotransposon that were similar to that of small HSP genes, thus explaining the overexpression of this transposable element under thermal stress. This work was by Egue F. et al. Phycologia 2015, 54 (6), 617–627 (DOI: 10.2216/15-52.1).

5) Line 143, “Moreover, under stress TEs can be activated…” (change place of “can”).

Author Response

We would like to thank you for the helpful comments and suggestions that improved our MS. We have considered all your suggestions raised, highlighted in yellow in the manuscript.

Point 1:

This word has been replaced (Ln 13).

Point 2:

The suggested reference has been added in (Ln 84).

Point 3:

The reference was corrected as suggested (third ref. on pg 12).

Point 4:

The recommended reference was added in (Ln 150).

Point 5:

The correction was made (Ln 163).

Reviewer 2 Report

In this manuscript titled “Novel insights into plant genome evolution and adaptation as revealed through transposable elements and non-coding RNAs in conifers”, the author pointed out the roles of transposable elements (TEs) and non-coding RNAs (ncRNAs) in the plant genome evolution by comparing the conifer genomes to the others. The major roles of transposable elements (TEs) and non-coding RNAs (ncRNAs) in the plant and the recent conifer genomic studies have been highly summarized.

The points in this review manuscript about TE and conifers are interesting, and the paragraphs are well written.

1.    There are some conifer TE database should be worth considering mentioned or included, such as ConTEdb;

2.    It might also be helpful to explain how to answer the “outstanding questions” based on our current knowledge;

3.    The points and evidence of ncRNAs are comparatively vague, in contract to those of TEs. It sounds to me the ncRNAs mentioned in this manuscript could include long ncRNAs which are originated from TEs, and most importantly the heterochromatic small interfering RNAs (hc-siRNAs) which suppress activity of TE. I don’t understand why the authors wouldn’t specify the biology of ncRNAs?

Author Response

We would like to thank you for the helpful comments and suggestions that improved our MS. We have considered all your suggestions raised, highlighted in yellow in the manuscript.

Point 1:

Thank you for providing this very pertinent information. Info has been added in (Ln 230).

Point 2:

Possible solutions to Outstanding Questions have been provided (pg 8).

Point 3:

This is a necessary clarification (pg 2).